# Individual/Joint Deblurring and Low-Light Image Enhancement in One Go via Unsupervised Deblurring Paradigm

## Abstract

Image restoration and enhancement, e.g., image deblurring, deraining and low-light image enhancement (LLIE), aim to improve the visual quality according to the corrupted/low-quality observation. Deep learning-based methods have achieved remarkable results on these individual tasks, but it is still hard to tackle them together. While some attempts have been made to implement joint task processing, they inevitably lead to higher data cost and higher training cost. Moreover, these attempts are strictly limited by the data distribution, i.e., the distribution of the inference data is required to be as close as possible to the training data, otherwise the data cannot be used for inference. In this paper, we take the LLIE and deblurring task as the subjects of this study in an attempt to seek a novel solution to the joint task processing problem. Specifically, we tackle this kind of problem in an extraordinary manner, i.e., *Individual/Joint Deblurring and Low-Light Image Enhancement in One Go via Unsupervised Deblurring Paradigm (DEvUDP)*, which integrates the noise self-regression and could avoid the limitations of aforementioned attempts. More specifically, a novel architecture with a transformation branch and a self-regression branch is elaborated, which only accepts unpaired blurry-sharp data as input to train the model; in this way, the pre-trained model can be surprisingly applied to both LLIE, deblurring and mixed degradation processing. Besides, we can choose to highlight perceptual performance or distortion performance of the model by configuring different components to the architecture. Extensive experiments have demonstrate the superiority of the method on different widely-used datasets.

## 1 Introduction

Image restoration and enhancement (IRAE) is a task for enhancing the visual quality according to a corrupted/low-quality observation with various degradations (e.g., rain, blur and low-light) and finally reconstructing the clear one. Benefiting from the powerful learning ability, deep learning-based IRAE techniques have made a great progress in obtaining high-quality and visually appealing results, e.g., deep learning-based image deblurring (Zhao et al., 2022b;a; Nah et al., 2017), deraining (Wei et al., 2021; Chen et al., 2022b), dehazing (Qin et al., 2020; Song et al., 2023), denoising (Lehtinen et al., 2018; Wang et al., 2022a) and low-light image enhancement (LLIE) (Wei et al., 2018; Zhang et al., 2022b; Guo et al., 2020; Li et al., 2022b).

Compared to tackle these IRAE tasks individually, joint task processing has a broader application scenario. However, the mainstream deep IRAE approaches are dedicated to tackle individual tasks due to greater challenges caused by the different distributions of degradations, and few approaches attempt to tackle them together. The existing joint task processing strategies can be roughly divided into three categories: 1) one-degradation-per-image (ODPI); 2) All-in-one (AiO); 3) multiple-degradations-per-image (MDPI). The comparison of these kinds of strategies are illustrated in Fig.1.

**ODPI-based** methods (see Fig.1 (a)) train the model independently for each individual degradation using the same model structure, and different pre-trained models are applicable for the corresponding degradation (Chen et al., 2021; Zamir et al., 2022; Tu et al., 2022; Wang et al., 2022b; Chen et al., 2022a). This kind of methods, e.g., MPRNet (Zamir et al., 2021), SwinIR (Liang et al., 2021) and GRL (Li et al., 2023), tackle different tasks one-by-one, which allows them to achieve impressive performance for each individual task. **AiO-based** methods (see Fig.1 (b)) use one model to

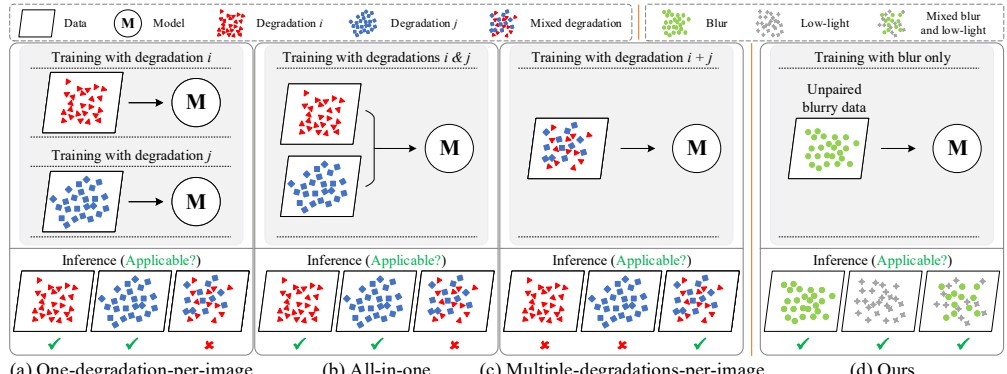

Figure 1: Comparison of joint task processing strategies. (a) One-degradation-per-image (ODPI): Train the model independently for each individual degradation using the same model structure, and the pre-trained model is applicable for the corresponding degradation. (b) All-in-one (AiO): Train one model using all data with different types of degradations, and the pre-trained model is applicable for multiple degradations. (c) Multiple-degradations-per-image (MDPI): Train one model using the data with mixed degradations, and the pre-trained model is only applicable for the mixed degradation. (d) Ours: Train one model with blur data only, while the pre-trained model is applicable for handling the cases of blur, low-light and mixed blur & low-light.

tackle all individual IRAE tasks (Li et al., 2022a). Specifically, to ensure all individual degradations are applicable for inference, this kind of methods, e.g., AirNet (Li et al., 2022a), train the model on the collection of all data with various degradations. However, for a given set of inference data, the AiO-based methods usually perform worse than the ODPI-based methods, since the distribution of the training data for the latter is closer to that of the inference data. **MDPI-based** methods (see Fig.1 (c)) trains one model using the data with mixed degradation for IRAE (Zhang et al., 2018b; Zhao et al., 2022c; Wan et al., 2022; Zhang et al., 2023). This kind of methods, e.g., FSGN (Song et al., 2019), RLED-Net (Ren et al., 2022) and LEDNet (Zhou et al., 2022), first construct the necessary datasets for training, based on which they design specific deep networks to handle various mixed degradation. Since the degradation of the training data is mixed, the pre-trained model is only applicable for those mixed degradations.

Although the aforementioned methods allow for joint task processing, they still have several limitations: 1) **higher data cost and higher training cost**; 2) **strict data distribution limitation**, which are determined by the processing strategy itself and are hard to avoid. For the limitation one, the ODPI and AiO strategies require ultra-large scale data support, which also results in a significant increase in training time, and in addition, the mixed degradation data required by MDPI is more difficult to collect compared to the data with individual degradation. For the limitation two, the distribution of the inference data is required to be as close as possible to training data, which means that the ODPI-based and AiO-based methods are hard to handle the mixed degradation, while the MDPI-based methods are hard to process various individual degradations.

To this end, we present a novel strategy for joint task processing without above limitations. Taking deblurring and LLIE tasks as the subjects of the study, we propose to deal with deblurring and LLIE individually and jointly in one go, called DEvUDP through an unsupervised deblurring paradigm[1] (see Fig.1 (d)). Specifically, we only use the deblur-related data (without LLIE-related data) for training and do the following tasks during inference: deblurring, LLIE and mixed degradation processing. Intuitively, it is difficult to be accomplished as the training data does not contain any low-light degradation, which results in a large difference in the distribution of the inference and training data. Fortunately, NoiSER (Zhang et al., 2022a) offers a straightforward solution to achieve LLIE without any low-light data, which makes our proposed strategy possible.

We summarize the main contributions of this paper as follows:

---

[1]A whole set of elements and operations. For example, the deblurring paradigm requires that all data and losses used must be deblur-related, and that other data (e.g., low-light image data) and losses are not allowed.

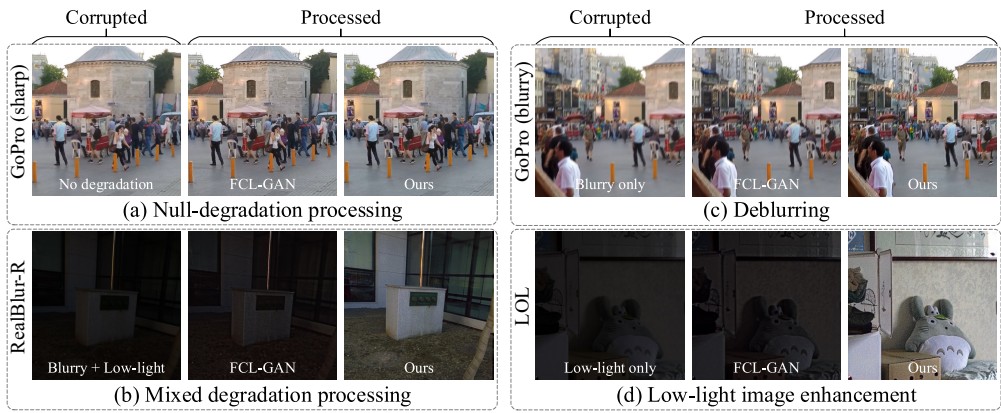

Figure 2: Visualization comparison of the processed results between current unsupervised deblurring SOTA method FCL-GAN (Zhao et al., 2022a) and our method.

- We propose a new strategy to jointly accomplish deblurring and LLIE. Specifically, we use a special approach to solve such joint task processing problem, i.e., only using an unsupervised deblurring paradigm for individual/joint deblurring and LLIE in one go, which can also reduce the data collection cost and model training cost, and alleviate the usage limitations arising from the differences in training-inference data distribution.

- We explicitly present a specific pipeline for using the unsupervised deblurring paradigm to enhance low-light images, during which the image deblurring itself and mixed degradation processing can also be achieved.

- We offer two modes so as to highlight the perceptual performance or the distortion performance by configuring different components to the model and keeping the entire architecture unchanged. Extensive experiments demonstrate the superiority of our approach.

## 2 RELATED WORK

### 2.1 UNSUPERVISED DEBLURRING METHODS

Unsupervised deblurring methods (Zhao et al., 2022b;a; Zhu et al., 2017; Yi et al., 2017; Lu et al., 2019) aim to transform a blurry image to a sharp one according to a series of unpaired data. Comparing to those supervised methods, they greatly decrease the data cost for training. UID-GAN (Lu et al., 2019) decomposes a blurry image into sharp content information and blur attributes, based on which a disentangled architecture is developed to deblur the image in face/text domain. FCL-GAN (Zhao et al., 2022a) introduces frequency-domain contrastive learning to develop a new lightweight and efficient unsupervised deblurring baseline that achieves advanced performance. These above methods are based on generative adversarial networks (GAN) (Goodfellow et al., 2014), while our approach is also based on this mechanism.

### 2.2 ZERO-REFERENCE LLIE METHODS

Zero-Reference LLIE methods (Guo et al., 2020; Li et al., 2022b; Liu et al., 2021; Ma et al., 2022) perform LLIE only using single low-light images without any paired/unpaired reference. Zero-DCE (Guo et al., 2020) and Zero-DCE++ (Li et al., 2022b) first introduce the concept of zero-reference to enhance low-light images. They explicitly transform the LLIE into the deep curve estimation problem and perform a pixel-to-pixel map using estimated deep curve to achieve LLIE. Inspired by retinex theory (Land, 1977), RUAS (Liu et al., 2021) establishes a network by unrolling the process of optimization and uses neural architecture search (NAS) (Liu et al., 2019) to compute the gradients of the architecture and parameters. As these methods are closer (comparing to paired/unpaired data-based methods) to ours at the level of data support, we choose them as a part of experiments.

### 2.3 NOISE SELF-REGRESSION TECHNIQUE

Self-regression (Hinton & Salakhutdinov, 2006; Kingma & Welling, 2014; Ulyanov et al., 2018) aims to use the input itself as supervised signal to reconstruct the output with similar texture. Noise self-regression (NoiSER) (Zhang et al., 2022a) is a novel technique for LLIE. The core idea of NoiSER is trying to provide a simple and efficient solution to enhance low-light images without any

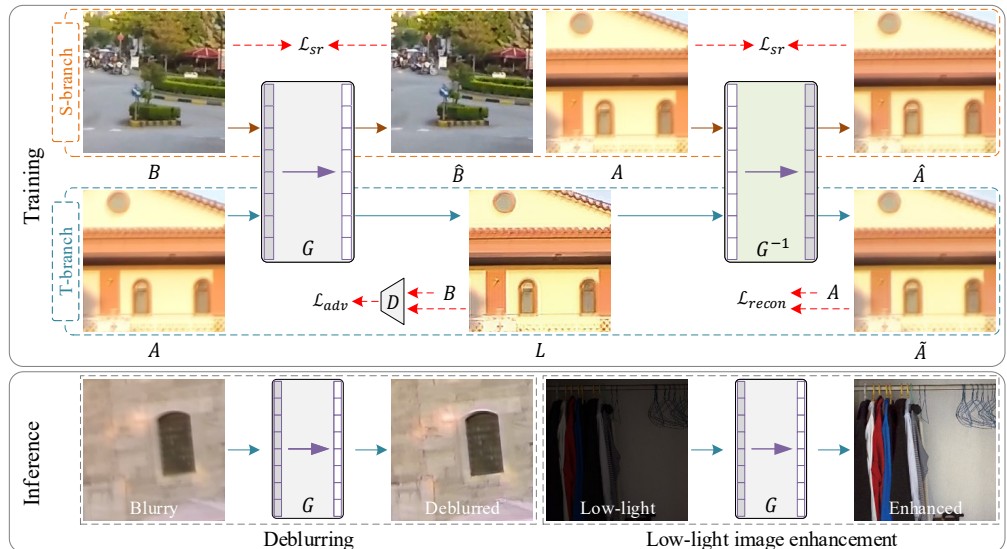

Figure 3: The architecture of DEvUDP which consists of two generators ($G$, $G^{-1}$) and one discriminator ($D$). During training, given unpaired images $A$ and $B$, the generator $G$ maps the blurry one to be sharp and $G^{-1}$ reverse this process (i.e., $A{\to}L{\to}\tilde{A}{\approx}A$); besides, self-regression operation is applied on $B$ and $A$ using generator $G$ and $G^{-1}$ respectively (i.e., $B{\to}\hat{B}{\approx}B$, $A{\to}\hat{A}{\approx}A$). The discriminator $D$ is used to determine the authenticity of the reconstructed image $L$. During inference, the pre-trained $G$ could perform deblurring, LLIE and mixed degradation processing in on go.

LLIE-related data. The basic pipeline is that: 1) randomly sampling the noise $n$ from the Gaussian distribution whose mean and variance is denoted by $\mu$ and $\sigma^2$; 2) developing a deep network with some instance normalization layers and without the global shortcut (He et al., 2016); 3) training the deep network using a certain loss imposed on the random noise and network output. The regression process can be expressed as follows:

$$\arg\min_{\theta} \mathbb{E}_n\{L(f_\theta(n), n)\}, \tag{1}$$

where $f_\theta$ is a parametric mapping family, $L$ denotes the loss function for minimizing the reconstruction error (here, $L_1$ is used) , and the value range of both $f_\theta(n)$ and $n$ is [-1,1].

## 3 PROPOSED METHOD

### 3.1 PROBLEM STATEMENT

Suppose we have a set of images with unknown lighting conditions and unknown blur conditions that need to be processed, what can we do to accomplish this task? Since the lighting and blur conditions are all unknown, there could be four possible combinations of the degradation in the above problem: ① no degradation, ② mixed degradation, ③ blur only, and ④ low-light only. Fig.1 has introduced the existing three strategies for joint task processing, however, it is obviously not applicable in this situation. There is still a solution to the above case (labeled by "AiO-like"): collecting a **hyper-scale** dataset containing the above four degradations, designing a **hyper-scale** deep model and then training the model for an **ultra-long** time. Nevertheless, the performance is not ensured, since AiO-like strategy brings a certain degree of performance decrease (Li et al., 2022a).

To this end, inspired by NoiSER technique (Zhang et al., 2022a), we attempt to develop a novel joint task processing strategy (DEvUDP) to deal with the above four degradations (degradation ① can be regarded as null-degradation [2]) with less data/training costs. In a word, we try to joint deblurring and LLIE only using unsupervised deblurring paradigm.

Fig.2 intuitively demonstrates the difference in visual effects between the general unsupervised deblurring method FCL-GAN (Zhao et al., 2022a) and ours. Fig.2 (b), Fig.2 (c) and Fig.2 (d) show the ability of our method to handle different degradations respectively. While Fig.2 (a) shows that our

---

[2]Null-degradation means that the image with null-degradation is equal to the image itself. The model is applicable to null-degradation means that the output of the model should be close to the input itself.

approach can handle null-degradation well, since our result is quite close to input itself. In the experiments, we do not provide additional experiments on null-degradation and only exemplify the ability of our approach to deal with null-degeneracy in Fig.2. Next, we detail how to perform this joint task processing.

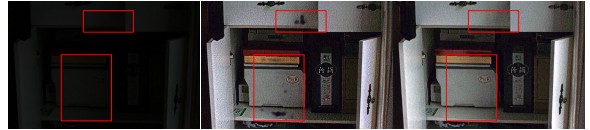

(a) Low-light    (b) T-branch only   (c) T&S-branches

Figure 4: Visualization comparison of the processed results with/without S-branch. As can be seen, S-branch helps to remove the artifacts and noise.

## 3.2 ARCHITECTURE

Let $\mathbb{A}$ and $\mathbb{B}$ denote two different domains respectively, i.e., blurry domain and sharp domain, and let $A \in \mathbb{A}$ and $B \in \mathbb{B}$ denote the images in the corresponding domain with different contents. We then present the whole joint deblurring and LLIE architecture for DEvUDP, as shown in "Training" part of Fig.3. From a structural point of view, the whole architecture consists of two generators $(G, G^{-1})$ and one discriminator $(D)$. The generator $G$ is used to map the blurry images to the sharp domain while the generator $G^{-1}$ reverses the process according to the output of $G$; the discriminator $D$ is used to identify whether an image in a domain is indeed belong to that domain according to a series of real/fake samples. From a functional point of view, the whole architecture is asymmetric and consists of two branches, i.e., Transformation branch (T-branch) and Self-regression branch (S-branch). These two branches are responsible for different goals, and one cannot be missing.

**T-branch**, as the most basic functional component of the architecture, aim to achieve a transformation of an image from blurry domain to sharp domain, in which the content of the image should be maintained. Specifically, given a blurry image $A$, we first use the generator $G$ to produce the latent sharp image $L$ according to $A$, which can be denoted by the following formula:

$$L = G(A). \tag{2}$$

To make the latent sharp image $L$ closer to sharp domain, given a sharp image $B$, we use LSGAN (Mao et al., 2017) to perform adversarial constraint on $L$, in which the latent $L$ is regarded as a fake sample while the image $B \in \mathbb{B}$ is regarded as a real one. The adversarial constraint is performed as

$$\mathcal{L}_D = \frac{1}{2}\mathbb{E}_{B\sim p(B)}\left[\left(D\left(B\right)-1\right)^2\right] + \frac{1}{2}\mathbb{E}_{L\sim p(L)}\left[D\left(L\right)^2\right], \tag{3}$$

$$\mathcal{L}_G = \mathbb{E}_{L\sim p(L)}\left[\left(D\left(L\right)-1\right)^2\right], \tag{4}$$

where $p(B)$ denotes the distribution of real samples and $p(L)$ denotes the distribution of fake samples. In the following we uniformly denote the adversarial loss by $\mathcal{L}_{adv}$.

To ensure that the content information of the latent sharp image $L$ and the blurry input $A$ are as consistent as possible, a suitable constraint should be imposed on $L$. It is not desirable to impose direct pixel-by-pixel/feature constraints between $L$ and $A$ as this would introduce a degree of blurry attributes. As a result, we introduce an auxiliary generator $G^{-1}$ to achieve the opposite process to the main generator $G$ and output the reconstructed image $\tilde{A}$ as:

$$\tilde{A} = G^{-1}(L). \tag{5}$$

After obtaining the reconstructed image $\tilde{A}$, we can indirectly retain the content information of $A$ without introducing any blurring attributes by imposing a pixel-by-pixel reconstruction constraint between $A$ and $\tilde{A}$. The corresponding constraint is as follows:

$$\mathcal{L}_{recon} = \|A - \tilde{A}\|_1, \tag{6}$$

where $\|\cdot\|_1$ is $L_1$ norm.

T-branch basically achieves the task of deblurring, however it does not allow for better low-light image enhancement. In fact, the T-branch can already adjust the brightness of the image to some extent, the reasons for which are threefold (Zhang et al., 2022a): (1) some Instance Normalization (IN) (Ulyanov et al., 2016) layers are contained in generator (see Section 3.3 for detail); (2) the mapping interval of the model is set to [-1,1] in which the median value of 0 falls within the normal-light range; (3) the process of $A \rightarrow \tilde{A} \approx A$ implicitly mimics the noise self-regression process.

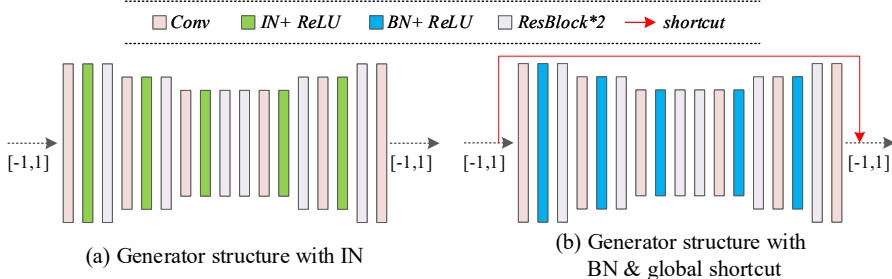

Figure 5: The different structures of the generator. The pixel range for both input and output is [-1,1]. The difference between (a) and (b) are the normalization and the shortcut.

Based on the above three points, following NoiSER (Zhang et al., 2022a), more direct reasons can be explained as: (i) the generator $G$ has the ability to reconstruct the contrast of adjacent pixels and produce the output that satisfies the gray-world hypothesis (Buchsbaum, 1980); (ii) the IN in the generator $G$ helps to remediate the overall brightness of the image. However, it does not guarantee the same pixel-level correspondence between the processed results and the model inputs as does self-regression, which leads to some obvious artifacts and noise in the processed results, as shown in Fig.4 (a) and Fig.4 (b). Thus, we introduce the S-branch to implement self-regression explicitly, rather than mimicking it implicitly as in reason (3). After introducing S-branch, noise and artifacts are significantly reduced, as shown in Fig.4 (a) and Fig.4 (c).

**S-branch** aims to alleviate noise and artifacts and to improve the visual quality of processed images. In T-branch, the main generator $G$ is used to implement the mapping from blurry domain to sharp domain, i.e., $G$: $\mathbb{A} \to \mathbb{B}$, and the auxiliary generator $G^{-1}$ is used for implementing the opposite mapping, i.e., $G^{-1}$: $\mathbb{B} \to \mathbb{A}$. Thus, performing self-regression on $G$ with blurry images, i.e., $G$: $\mathbb{A} \to \mathbb{A}$, is in conflict with the function of $G$ itself, which means that when given a blurry image $A \in \mathbb{A}$, the generator $G$ will have no idea which domain to map $A$ to.

To successfully perform self-regression operations on $G$ while avoiding the introduction of additional conflicts, we use sharp images $B \in \mathbb{B}$ to implement the self-regression on $G$, i.e., $G$: $\mathbb{B} \to \mathbb{B}$. This is also the key to our model's ability to maintain the content of the input when a sharp image is given (see Fig.2 (a)). Specifically, given a sharp image $B \in \mathbb{B}$, the generator $G$ is forwarded to obtain the corresponding output $\hat{B}$ as follows:

$$\hat{B} = G(B). \tag{7}$$

Then we could impose the following constraint to perform self-regression operation on $G$:

$$\mathcal{L}_{sr}^{\mathbb{B}} = \|B - \hat{B}\|_1. \tag{8}$$

As the auxiliary generator $G^{-1}$ could provide posteriori information to $G$, it is also necessary to perform self-regression on $G^{-1}$ using the images in blurry domain, i.e., $G^{-1}$: $\mathbb{A} \to \mathbb{A}$. Given a blurry image $A \in \mathbb{A}$, we forward $G^{-1}$ to obtain $\hat{A}$ and perform self-regression constraint on $G^{-1}$ as

$$\hat{A} = G^{-1}(A), \mathcal{L}_{sr}^{\mathbb{A}} = \|A - \hat{A}\|_1. \tag{9}$$

**Co-training and Inference.** In order to make T-branch and S-branch cooperate better, we impose the following total constraints on the architecture for DEvUDP:

$$\mathcal{L}_{total} = \mathcal{L}_{adv} + \lambda_{recon}\mathcal{L}_{recon} + \lambda_{sr}(\mathcal{L}_{sr}^{\mathbb{A}} + \mathcal{L}_{sr}^{\mathbb{B}}), \tag{10}$$

where $\lambda_{recon}$ and $\lambda_{sr}$ are tunable parameters, which are set to 10 and 5 respectively.

From the above, it is clear that our training fully satisfies the unsupervised deblurring paradigm, i.e., 1) we only use unpaired blurry-sharp images for training and do not use any other data; 2) the model we used belongs to the general deep neural network model; 3) we only impose regular adversarial constraints and $L_1$ norm on the architecture and no other constraints (e.g. exposure control constraint (Guo et al., 2020; Li et al., 2022b) specifically for LLIE). Despite this, during inference, **the pretrained model can handle blur, low-light and mixed degradation in one go**, which is attributed to the clever utilization of NoiSER (Zhang et al., 2022a) technique to our approach.

Table 1: Quantitative comparison of deblurring performance on GoPro (Nah et al., 2017) and RealBlur-J (Rim et al., 2020) datasets. ∗ indicates that the image is scaled down before inference (640×360 for GoPro and 320×360 for RealBlur-J), which is because CRNet(Zhao et al., 2022b) is very demanding on GPU memory. IT denotes the inference time in milliseconds (GPU: 2080Ti; resolution: 1280×720); No.P denotes the number of parameters for inference. Clearly, our method outperforms other methods in most metrics.

| Methods | GoPro | | | | RealBlur-J | | | | IT↓ | No.P↓ |
|---|---|---|---|---|---|---|---|---|---|---|
| | PSNR↑ | SSIM↑ | MAE↓ | LPIPS↓ | PSNR↑ | SSIM↑ | MAE↓ | LPIPS↓ | | |
| CycleGAN | 22.54 | 0.720 | 13.95 | 0.243 | 19.79 | 0.633 | 22.39 | 0.247 | 8.5 | 11.38M |
| UID-GAN | 23.56 | 0.738 | 11.38 | 0.289 | 22.87 | 0.671 | 13.41 | 0.235 | 64.3 | 19.93M |
| FCL-GAN | 24.84 | 0.771 | 9.63 | 0.239 | 25.36 | 0.736 | 9.58 | 0.193 | 14.1 | 2.77M |
| DEvUDP (Ours) | 22.19 | 0.708 | 14.30 | 0.253 | 19.75 | 0.617 | 22.18 | 0.261 | 6.4 | 6.95M |
| DEvUDP-DT (Ours) | 25.15 | 0.786 | 9.56 | 0.272 | 25.87 | 0.790 | 8.69 | 0.201 | 5.8 | 6.96M |
| CRNet* | 25.88 | 0.803 | 9.12 | 0.192 | 24.05 | 0.715 | 13.02 | 0.146 | - | 11.66M |
| DEvUDP-DT* (Ours) | 26.69 | 0.822 | 8.54 | 0.205 | 26.55 | 0.831 | 8.80 | 0.137 | - | 6.96M |

Finally, there is an interesting question: looking at Fig.3, assuming you already know that the normalization layer and the mapping interval of the generator $G$ are IN and [-1,1] respectively, and that $G$ has no global shortcut, can you imagine the pre-trained $G$ possesses such interesting functions if you had never read this paper? ***This is the revelation brought by our approach.***

Table 2: Quantitative comparison of LLIE performance on LOL (Wei et al., 2018) dataset.

| Methods | PSNR↑ | SSIM↑ | MAE↓ | LPIPS↓ |
|---|---|---|---|---|
| LIME | 14.22 | 0.514 | 50.92 | 0.368 |
| Zhang et al. | 14.02 | 0.513 | 51.91 | 0.372 |
| Zero-DCE | 14.97 | 0.500 | 41.62 | 0.432 |
| Zero-DCE++ | 14.80 | 0.516 | 43.72 | 0.413 |
| RUAS | 16.40 | 0.500 | 39.11 | 0.270 |
| SCI | 14.02 | 0.508 | 52.36 | 0.388 |
| DEvUDP (Ours) | 16.88 | 0.667 | 36.27 | 0.326 |

### 3.3 MODEL STRUCTURE

Following the design of the LED structure (Zhao et al., 2022a), we present the structure of $G$ in Fig.5 (a). Please note that this structure satisfies the following three conditions: 1) no global shortcut; 2) IN is used instead of other normalization layers; 3) the pixel range for both input and output is [-1,1], which is also the necessary conditions for the whole architecture to be able to handle low-light data.

Our proposed DEvUDP is based on an unsupervised deblurring paradigm. However, due to the introduction of IN, the distortion performance[3] for unsupervised deblurring will be at a lower level (Zhao et al., 2022a). Actually, in IRAE task, poor distortion performance does not mean poor perceptual performance[4], and vice versa. We find that using IN gives better perceptual performance than not using IN. Besides, to take the distortion performance of DEvUDP into account as well, we provide an alternative to construct the model with SOTA deblurring distortion performance while not having the ability for enhancement (denoted as DEvUDP-DT), as shown in Fig.5 (b).

### 3.4 ENHANCEMENT PIPELINE

We explicitly present the enhancement pipeline using unsupervised deblurring paradigm as follows:

1. Using unpaired blurry-sharp images for training, and the images should be of normal-light.

2. Downsizing the image while keeping the width-high ratio unchanged before all data augment, which can better mimic the "noise" data in NoiSER (Zhang et al., 2022a) to obtain a better visual effect (see ablation studies in supplementary materials for details).

3. Mapping the image's value range to [-1,1] in which the median 0 falls in normal-light range.

4. Preparing the generator according to Fig.5 (a).

5. Constructing and training the architecture displayed by Fig.3 according to the data in step (3) and the generator in step (4).

6. Enhancing the images according the pre-trained model obtained by step (5).

---

[3] Distortion performance refers to how much the algorithm distorts the input image.

[4] Perceptual performance refers to how well the algorithm performs against the human-eye visual effect.

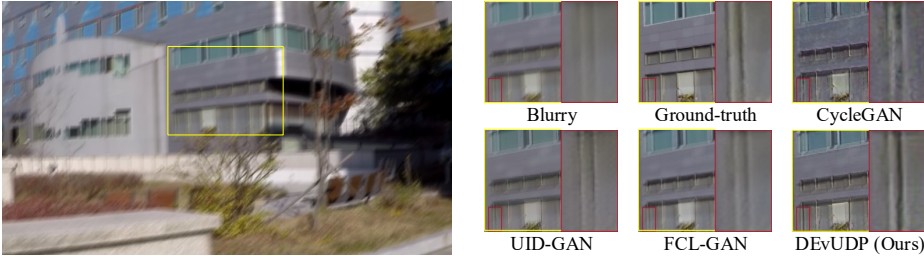

Figure 6: Visualization comparison on GoPro (Nah et al., 2017) dataset with several deblurring methods. Our DEvUDP achieves better deblurring effect meanwhile avoiding obvious artifacts.

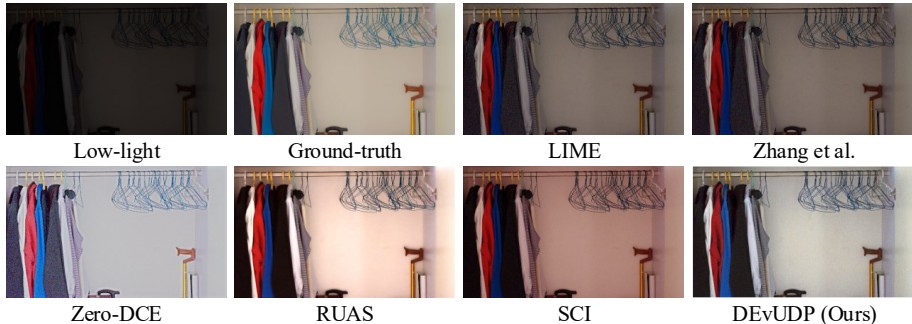

Figure 7: Visualization comparison on LOL (Wei et al., 2018) dataset with several LLIE methods. Clearly, our model achieves more natural and accurate enhancement results.

## 4 EXPERIMENTS

### 4.1 EXPERIMENTAL SETTINGS

**Datasets and evaluation metrics.** We use four datasets to evaluate the performance of each method, i.e., GoPro (Nah et al., 2017) and RealBlur-J (Rim et al., 2020) for deblurring, LOL (Wei et al., 2018) for LLIE and LOL-Blur (Zhou et al., 2022) for mixed degradation processing. We use four most widely-used metrics (i.e., PSNR, SSIM (Wang et al., 2004), MAE and LPIPS (Zhang et al., 2018a)) to measure the quantitative performance of each method. In experiments, ↑ means the higher the better and ↓ means the opposite; the best is marked in red and the second best is marked in blue.

**Compared methods.** We include ten methods for performance comparison, including four unsupervised deblurring methods (i.e., CycleGAN (Zhu et al., 2017), UID-GAN (Lu et al., 2019), CRNet (Zhao et al., 2022b) and FCL-GAN (Zhao et al., 2022a)), and six traditional/zero-reference database LLIE methods (i.e., LIME (Guo, 2016), Zhang et al. (Zhang et al., 2019), Zero-DCE (Guo et al., 2020), Zero-DCE++ (Li et al., 2022b), RUAS (Liu et al., 2021) and SCI (Ma et al., 2022)).

**Implementation details.** The proposed architecture is implemented based on PyTorch 1.10 (Paszke et al., 2019) and NVIDIA A100 with 40G memory. We set 200 epochs for training with batch size 4, using Adam (Kingma & Ba, 2015) with $\beta_1$=0.5 and $\beta_2$=0.999 for optimization. The initial learning rate was set to 0.0002, which was reduced by half every 50 epochs. For data augment, we first randomly crop the image to 256×256 and then perform horizontal flip with probability 0.5. During inference, only the main generator $G$ is used to transform the given input, as denoted by Eqn.2.

### 4.2 EXPERIMENTAL RESULTS

Since the ability of our DEvUDP for handling the degradation is threefold, i.e., 1) deblurring; 2) LLIE; 3) mixed degradation processing, we conduct experiments on each of the above three tasks. The experimental results and evaluations are shown in the following.

**(1) Experimental results for deblurring.** We train our DEvUDP on the training set of GoPro (Nah et al., 2017) and evaluate the fitting and generalization abilities on GoPro and RealBlur-J (Rim et al., 2020), respectively. The quantitative results are shown in Table 1. As we can see, our DEvUDP outperforms other methods in most metrics, which demonstrates that our model has stronger fitting and generalization abilities for deblurring. Fig.6 shows the visualization results of our DEvUDP and other methods, from which we can see that the result of our DEvUDP is more visual pleasing.

Table 3: Quantitative comparison of mixed degradation processing on LOL-Blur (Zhou et al., 2022).

| Methods | Direction | PSNR↑ | SSIM↑ | MAE↓ | LPIPS↓ |
|---|---|---|---|---|---|
| FCL-GAN → Zero-DCE | Deblurring→Enhancement | 9.07 | 0.515 | 86.03 | 0.495 |
| Zero-DCE → UID-GAN | Enhancement→Deblurring | 12.23 | 0.574 | 59.46 | 0.368 |
| UID-GAN → RUAS | Deblurring→Enhancement | 6.88 | 0.454 | 109.46 | 0.510 |
| RUAS → FCL-GAN | Enhancement→Deblurring | 12.72 | 0.602 | 57.50 | 0.406 |
| DEvUDP (Ours) | - | 18.03 | 0.653 | 28.14 | 0.323 |

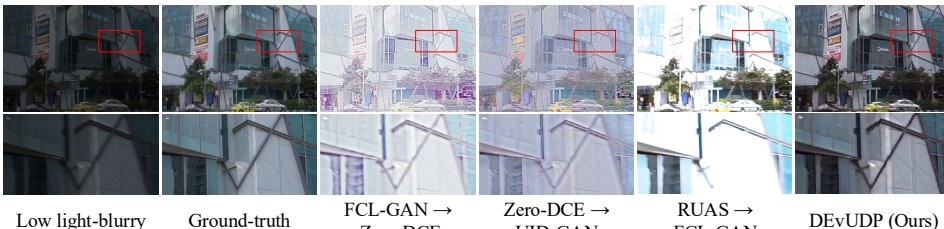

| Low light-blurry | Ground-truth | FCL-GAN → Zero-DCE | Zero-DCE → UID-GAN | RUAS → FCL-GAN | DEvUDP (Ours) |

Figure 8: Visualization comparison on LOL-Blur (Zhou et al., 2022) dataset. Six groups of results are displayed, with the bottom of each group indicating a zoom in on the content of the red box at the top. Clearly, our DEvUDP has the ability to process mixed degradation better.

**(2) Experimental results for LLIE.** We directly apply the model pre-trained on the deblurring dataset GoPro (Nah et al., 2017) to enhance low-light images. Table 2 and Fig.7 compare the quantitative and visualization results with several methods on widely-used LOL (Wei et al., 2018) dataset. We see that our method achieves superior numerical results and more natural visual effect, compared to other traditional/zero-reference data-based methods. That is, DEvUDP possesses the ability to enhance low-light images, although it is pre-trained by pure unsupervised deblurring paradigm.

**(3) Experimental results for mixed degradation processing.** Since no unsupervised method has been developed to process mixed degradation, we conduct experiments for mixed degradation processing by the following two forms: ① deblurring → enhancement (D→E), ② enhancement → deblurring (E→D). Specifically, we choose two SOTA unsupervised deblurring methods (i.e., FCL-GAN (Zhao et al., 2022a) and UID-GAN) and two SOTA zero-reference LLIE methods (i.e., Zero-DCE (Guo et al., 2020) and RUAS (Liu et al., 2021)) to cross-construct a experimental ring (i.e., FCL-GAN → Zero-DCE → UID-GAN → RUAS → FCL-GAN), from which two D→E methods and two E→D methods are constructed.

We directly apply the model pre-trained on the deblurring dataset GoPro (Nah et al., 2017) to process mixed degradation on LOL-Blur (Zhou et al., 2022). Table 3 shows the quantitative comparison with the above methods, and clearly, the quantitative results of our DEvUDP are higher than other methods in all metrics. Fig.8 demonstrates the visual comparison for processing mixed degradation, from which we can see that: 1) from the deblurring perspective, our DEvUDP has stronger ability than other methods for deblurring, since the processed result is sharper; 2) from the LLIE perspective, our DEvUDP could obtain more natural results than other related methods, since the result is more close to ground-truth. So far, we also prove that our DEvUDP has the ability for mixed degradation processing, although it is pre-trained by a pure unsupervised deblurring paradigm.

## 5 CONCLUSION

In this paper, we present a novel strategy to perform individual and joint task processing of deblurring & low-light image enhancement in one go via an unsupervised deblurring paradigm. That is, we train the model with only unpaired deblur-related data, which seems outside of general cognition. However, it does work and the model pre-trained with this strategy has the ability to handle blur, low-light and mixed degradation in one go. The reason why this strategy works is the embedding of self-regression technique throughout the whole unsupervised deblurring process, and then we explicitly present the detailed pipeline for using this strategy to enhance low-light images. In addition, to consider the distortion performance and perceptual performance simultaneously, the proposed model is allowed to emphasize the required performance by equipping with different components. Extensive experiments for individual/joint deblurring and LLIE tasks have verified the superiority of our proposed strategy. Moreover, we also demonstrate the ability of our strategy for other joint task processing, e.g., individual/joint deraining and LLIE, which can be found in supplementary.

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
