# OpenReview forum: "Individual/Joint Deblurring and Low-Light Image Enhancement in One Go via Unsupervised Deblurring Paradigm"
_ICLR.cc/2024/Conference — ICLR 2024 Conference Withdrawn Submission_

### Official Review · Reviewer_q5w6 · 2023-10-30

**Soundness:** 3 good
**Presentation:** 2 fair
**Contribution:** 2 fair
**Rating:** 3
**Confidence:** 4

**Summary:**

In this paper, the authors introduce an unsupervised deep learning method for simultaneous individual and joint deblurring and low-light image enhancement. The crux of their approach is rooted in an insight from NoiSER (Zhang et al., 2022a) that suggests the potential for low-light image enhancement through an instance-normalization layer, employing random noise images as both input and output for each pixel within training pairs. Consequently, this approach seamlessly integrates low-light image enhancement within the deblurring framework. The efficacy of this approach is assessed through experiments conducted under combined settings.

**Strengths:**

1. Simultaneous deblurring and low-light image enhancement present an intriguing yet formidable challenge.

2. The suggested method is both unsupervised and tackles both tasks concurrently.

**Weaknesses:**

1. While this paper ambitiously addresses joint task processing, as outlined in the Introduction, it primarily focuses on a single integrated task—joint deblurring and low-light image enhancement. Consequently, the paper may exaggerate the breadth of its application scope.

2. Given this broad scope, the paper lacks an in-depth analysis of the specific task it tackles. Notably absent are discussions on the predominant types of blur in low-light conditions, the scenarios where low-light enhancement and deblurring coincide, and the characteristic features of blurred low-light images. These omissions hinder the provision of insights into low-light deblurring scenarios.

3. The novelty of the paper appears somewhat limited, primarily hinging on the utilization of NoiSER (Zhang et al., 2022a) and normalization layers. Furthermore, the GAN-based techniques employed for unsupervised deblurring are relatively commonplace.

**Questions:**

1. What specific type of blur is implied within the paper's context?

2. Is it possible for the authors to provide a thorough discussion of the image formation model in low-light environments, particularly concerning the associated blurring effects?

---

### Official Review · Reviewer_3D3W · 2023-10-30

**Soundness:** 2 fair
**Presentation:** 2 fair
**Contribution:** 2 fair
**Rating:** 3
**Confidence:** 5

**Summary:**

This paper solved the problem of simultaneous deblurring and low-light enhancement using an unsupervised deblurring approach. The deblurring is mainly tackled by a GAN network, while the low-light enhancement is done by the well-established NoiSER technique. Experiments on lowlight-degraded GoPro and blurred LOL dataset are conducted for performance evaluation.

**Strengths:**

+ Joint processing is of additional values in comparison to tradditional separate treatment.
+ The proposed method is done in one go and does not rely on paired data.

**Weaknesses:**

- Only synthetic data is used for training/test. The effectiveness of training and test on real-world scenarios of simultaneous blurry and low-light degradations remains unkown. This is particularly important for an unsupervised learning method.
- The main contribution is the smart exploitatoin of the existing NoiSER tehcnique, which is insufficient for the high standards of ICLR.
- The blur models assumpted in low-light enviroments are not metioned at all.

**Questions:**

See the weakenss part.

---

### Official Review · Reviewer_Vy5q · 2023-10-30

**Soundness:** 2 fair
**Presentation:** 3 good
**Contribution:** 2 fair
**Rating:** 5
**Confidence:** 4

**Summary:**

This paper takes the low-light image enhancement and deblurring task as the subjects in this study in an attempt to seek a novel solution to the joint task processing problem. It designs a novel architecture with a transformation branch and a self-regression branch is elaborated. It only accepts unpaired blurry-sharp data as input to train the model. The pre-trained model can be surprisingly applied to both LLIE, deblurring and mixed degradation processing.

**Strengths:**

1. The differences between the existing three types of approaches and the proposed approach are clear. By training the model on data with only one type of degradation, it can be applied to handle two types of degradations and mixed degradations.
2. The proposed method can individually or jointly handle multiple types of degradations.
3. It shows superiority over the combination of existing low-light image enhancement and deblurring methods.

**Weaknesses:**

1. The method does not contain denoising-related operations. In Figs. 2 and 7, and the results in the supplementary materials, the noise of the enhanced result is obvious.
2. The proposed method is not designed to address the issue of LLIE. It seems like the proposed method is accidentally found to have this function during the experimental process. The function of IN for LLIE is reasonable as it can eliminate domain differences and the target domain is normal-light and sharp images. The second and third reasons are a bit farfetched.
3. It lacks the comparison with the comparison with the most relevant method:
LEDNet: Joint Low-Light Enhancement and Deblurring in the Dark

**Questions:**

1. For low-light images, their dynamic range may vary greatly in different scenes. The generalization of enhancement performance is always a challenge for various LLIE methods; For deblurring, differences in the size and blurring kernels can also lead to differences in the distributions of degraded data. The generalization is also a challenge. The proposed method not only considers multiple degradation methods, but also considers the diversity of data in each degradation method. Has the generalization in this regard been considered and how the generalization is ensured?
2. The method does not contain denoising-related operations and the blurred data in the training phase does not contain noise. Why does the method perform best in low-light image enhancement?

---

### Official Review · Reviewer_KNEC · 2023-10-31

**Soundness:** 3 good
**Presentation:** 3 good
**Contribution:** 3 good
**Rating:** 5
**Confidence:** 5

**Summary:**

This submission proposes an unsupervised image restoration approach, DEvUDP, that is trained on unpaired blur/sharp images. The DEvUDP, with a transformation branch and a self-regression branch, is elaborated to perform individual and joint task processing of deblurring and low-light image enhancement in one go via an unsupervised deblurring paradigm. Experiments on several datasets demonstrate the effectiveness of the proposed method when compared with the chosen baselines.

**Strengths:**

The strengths are:

1) Proposing an unsupervised approach that can handle image deblurring and low-light enhancement individually and jointly.

2) Achieving competitive performance compared with chosen baselines.

**Weaknesses:**

The weaknesses of this paper are listed as follows:

1) The introduction section is well organized and effectively outlines the improvements and advantages of the proposed method, while the other sections can be improved.

2) Please kindly check Eq. (3) and Eq. (4).

3) Unclear parts. It is unclear how the model has the ability of low-light image enhancement.

3.1) It would be better to demonstrate the effectiveness of NoiSER (Arxiv) before using it.

3.2) For consistency, it would be better to briefly introduce the main idea of NoiSER in the paper.

4) The results reported in Table 1 on the GoPro need to be checked or provide more details.
For example, UID-GAN is reported as 23.56dB in the submission while being 25.59dB in [a].

[a] Y. Wen et al., "Structure-Aware Motion Deblurring Using Multi-Adversarial Optimized CycleGAN," in IEEE Transactions on Image Processing, vol. 30, pp. 6142-6155, 2021, doi: 10.1109/TIP.2021.3092814.

5) It would be better to include more classical conventional deblurring algorithms as baselines. Those classical conventional deblurring algorithms do not need ground-truth sharp images as well.

6) It would be better also to explain whether weights shared the generator G and the auxiliary generator G^{−1}.

7) The sentence ‘We directly apply the model pre-trained on the deblurring dataset GoPro (Nah et al., 2017) to enhance low-light images’ is unclear to me. Does the pre-trained model follow the enhancement pipeline in section 3.4?

**Questions:**

Seeing Weaknesses.